# Surface Crystal Nucleation and Growth in Poly (ε-caprolactone): Atomic Force Microscopy Combined with Fast Scanning Chip Calorimetry

**DOI:** 10.3390/polym13122008

**Published:** 2021-06-19

**Authors:** Rui Zhang, Mengxue Du, Evgeny Zhuravlev, René Androsch, Christoph Schick

**Affiliations:** 1Centre CALOR, Institute of Physics and Competence, University of Rostock, 18051 Rostock, Germany; rui.zhang@chemie.uni-halle.de (R.Z.); evgeny.zhuravlev@uni-rostock.de (E.Z.); 2Interdisciplinary Center for Transfer-Oriented Research in Natural Sciences (IWE TFN), Martin Luther University Halle-Wittenberg, 06099 Halle/Saale, Germany; mengxue.du@iw.uni-halle.de (M.D.); rene.androsch@iw.uni-halle.de (R.A.); 3Key Laboratory of High-Performance Polymer Materials and Technology of Ministry of Education, and The State Key Laboratory of Coordination Chemistry, Department of Polymer Science and Engineering, School of Chemistry and Chemical Engineering, Nanjing University, Nanjing 210093, China; 4Shenyang Research Institute, Nanjing University, Shenyang 224300, China; 5Butlerov Institute of Chemistry, Kazan Federal University, 18 Kremlyovskaya Street, 420008 Kazan, Russia

**Keywords:** fast scanning chip calorimetry, atomic force microscopy (AFM), AFM-FSC combination, poly (ε-caprolactone) (PCL), surface and bulk crystal growth rate, banded spherulites

## Abstract

By using an atomic force microscope (AFM) coupled to a fast scanning chip calorimeter (FSC), AFM-tip induced crystal nucleation/crystallization in poly (ε-caprolactone) (PCL) has been studied at low melt-supercooling, that is, at a temperature typically not assessable for melt-crystallization studies. Nanogram-sized PCL was placed on the active/heatable area of the FSC chip, melted, and then rapidly cooled to 330 K, which is 13 K below the equilibrium melting temperature. Subsequent isothermal crystallization at this temperature was initiated by a soft-tapping AFM-tip nucleation event. Crystallization starting at such surface nucleus led to formation of a single spherulite within the FSC sample, as concluded from the radial symmetry of the observed morphology. The observed growth rate in the sub-micron thin FSC sample, nucleated at its surface, was found being much higher than in the case of bulk crystallization, emphasizing a different growth mechanism. Moreover, distinct banding/ring-like structures are observed, with the band period being less than 1 µm. After crystallization, the sample was melted for gaining information about the achieved crystallinity and the temperature range of melting, both being similar compared to much slower bulk crystallization at the same temperature but for a much longer time.

## 1. Introduction

The in-situ combination of an atomic force microscope (AFM) with a fast scanning chip calorimeter (FSC) enables the visualization of the nanometer-scale morphology of polymer samples subjected to specific thermal histories. In a recently developed device, the FSC sensor is directly placed into an AFM and used as a hot-stage [1], yielding several advantages compared to earlier attempts of assessing the structure of polymer samples prepared in an FSC. While in the past it was needed to destroy the FSC sensor holding the sample subjected to a well-defined thermal profile [2,3,4,5], with the new setup such sensor destruction is not required. This allows calorimetric analysis of the sample after imaging for further correlation of the structure with thermal properties assessable in the FSC heating scans, including the temperature and enthalpy of melting or the glass transition temperature. This way, as an example, it was possible to determine the enthalpy of crystallization/melting of a single crystal of polyamide 66 formed at a specific crystallization condition [6]. A further advantage of the new device is the possibility of repeated imaging of a selected sample/sample-spot exposed to specific thermal treatments, permitting e.g., following structure evolution as a function of time during a phase transformation. Worth noting that such experiments can be performed at any temperature and time scale as the short time constant and high cooling capacity of the FSC sensor/hot-stage [7] allows freezing all intermediate steps of phase transformations by cooling the system to below the glass transition temperature (*T*_g_) for AFM scanning. Such an approach has been applied to obtain the kinetics of homogeneous crystal nucleation of PA 66 in a wide range of temperatures [6]. Analysis of the kinetics and phenomenology of melting is a further research subject, which, however, has not been tested yet.

A different experimental route of observation of semicrystalline morphologies using the AFM-FSC device, besides quenching and freezing structures formed at elevated temperature to below *T*_g_ for subsequent AFM scanning, involves imaging of the sample surface at the temperature of the phase transformation, that is, above *T*_g_. First studies about stand-alone (that is, not coupled to a calorimeter) in-situ hot-stage-based AFM analysis of polymer crystallization date back to the 1990s and by now this tool has been applied for investigation of numerous polymers [8,9,10,11,12,13,14,15,16,17,18]. In these studies, the sample temperature was adjusted by using a hot-stage while scanning the surface with a non-actively heated tip of unknown temperature, in both, contact- or non-contact modes. However, it was early recognized that such setup introduces uncertainty regarding the true surface temperature caused by undefined heat flow from the sample towards the tip, even if being detached from the sample [19]. Perhaps even more important than a constant offset between measured hot-stage temperature and true sample-surface temperature, is the strong supercooling of the tip-sample contact area, eventually causing crystal nucleation [20]. In addition, in particular when the AFM is operated in contact mode, there may be induced deformation/shear of the (molten) sample causing additional nucleation effects [14,16,21]. However, regardless the specific interaction of the AFM tip with the polymer surface including nucleation effects to initiate a crystallization process, in-situ AFM analysis of the melt-crystallization process provides valuable information about mechanisms of growth and formation of specific crystal morphologies. In the present work, this tool is applied for studying AFM-tip-nucleated crystallization of poly (ε-caprolactone) (PCL) with the attached FSC serving for assuring a well-defined thermal pathway when approaching the analysis temperature and for evaluation of the melting behavior and achieved crystallinity after the crystallization process.

PCL is an aliphatic, biodegradable, and crystallizable polymer with many applications in the medical sector [22,23,24] or as shape-memory material [25]. The equilibrium melting temperature (*T*_m,0_) and glass transition temperature (*T*_g_) are slightly higher than 343 K [26,27,28,29] and around 203 K [30], respectively, and the maximum crystallinity is around 50% [31]. The kinetics of the crystallization behavior has been studied in detail, revealing that crystallization is fastest slightly below 0 °C, with the crystallization half-time at this temperature being around 0.1 s [32,33]. Melt-crystallization typically proceeds via spherulitic growth of lamellae [33,34]. In-situ hot-stage AFM analyses of crystallization of thin films of PCL at different temperatures showed formation of, with respect to the substrate, flat-on oriented lozenge-shaped lamellae when crystallizing at low supercooling, with observed overgrowth above and below the basal lamellae [35]. At lower temperature, also hedrites (regular polygonal formations of lamellae [36]) and dendrites were observed [21,35]. These morphological studies are now further advanced by analysis of melt-crystallization of PCL at low supercooling, starting at a single surface nucleus.

## 2. Materials and Methods

We used a PCL grade from Sigma-Aldrich (Burlington, VT, USA) (Product-Number 440752-5G) with a mass-average molar mass and polydispersity of 20 kg/mol and 1.73, respectively [32,37], and an FSC-AFM instrumental setup, consisting of a Level AFM from Anfatec (Oelsnitz (V), Germany) [38] and a home-built FSC provided by Functional Materials e.V. Rostock (Rostock, Germany), described elsewhere [39,40]. In detail, the FSC was equipped with the XI395 sensor from Xensor Integration (EJ Delfgauw, The Netherlands) [41], and for AFM imaging, backside gold-coated Scout 70 RAu silicon probes from NuNano (Bristol, UK) with a resonance frequency of 70 kHz, tip radius of 5 nm, and spring constant of 2 N/m (0.5–3.5 N/m), respectively, were employed. The AFM was operated in non-contact mode, with simultaneous recording of the height profile and phase angle and amplitude of oscillations at a spatial resolution of 512 pixel × 512 pixel within about 1200 s/image. The entire system was purged with nitrogen gas.

Sample preparation included the dissolution of PCL in tetrahydrofuran with a purity > 99.9%, obtained from Sigma Aldrich (Burlington, VT, USA), for subsequent spin-coating of a thin film with a thickness of about 300 nm. Tetrahydrofuran has been proven to be an excellent solvent for PCL as being used for gel permeation chromatography (GPC) as well as it is not affecting the chemical structure of the polymer. The obtained film was cut to obtain a square with the size of about 25 × 25 μm^2^ which then was transferred to the center of the heatable area of the sensor using a thin copper wire. In order to obtain a flat and smooth film, the sample was first conditioned by heating to successively higher temperatures. In detail, the sample was heated at 10,000 K/s from 285 K to 320 K, annealed for 0.01 s, cooled at 10,000 K/s to 285 K, kept there for 0.01 s, and then the whole procedure was repeated by heating to a 10-K-higher temperature, until the final temperature of 400 K was reached. After that, the sample was repeatedly heated to 400 K and cooled to 285 K until the sample was stable from the point of view of the recorded heat-flow rate signal. Figure 1 shows the sample in the center of the heatable area of the FSC chip, revealing that the initial square turned into an ellipsoidal droplet-like shape, with the long and short dimensions of the ellipsoid being around 28 and 16 µm, respectively. The height and mass of the sample are estimated to 750 nm and 6 ng, respectively, with the latter derived by analysis of the absolute heat capacity in the liquid state, and comparing it with a mass-normalized reference value, available in the literature [42]. The maximum height of the droplet was measured by optical microscopy, by analyzing the depth of field.

After sample preparation, the AFM-tip was allowed approaching the sample at a location near its center, using an automatic procedure provided with the instrument’s software, assuring soft touching of the tip at the sample surface. Afterward, the tip was detached and vertically moved by a distance of 30 μm away from the sample surface. Then, the sample was heated to 400 K, to obtain an equilibrated melt by annealing for 0.01 s, followed by rapid cooling to 330 K at a rate of 100,000 K/s. During subsequent annealing of the metastable melt at 330 K [≈ *T*_m,0_–13 K] [42], the tip was lowered towards the sample surface for nucleating the crystallization process by a brief contact. After moving the tip vertically back to its former position, 30 µm above the sample, the cantilever resonance frequency was re-calibrated, confirming the absence of contamination of the tip as a prerequisite for subsequent mapping the structure. Re-approaching of the sample surface by the AFM-tip lasted 600 s, and during this time, crystallization occurred. As such, it was not possible to follow crystal growth but only detection of the final structure.

Complementary experiments with the goal to estimate the crystal growth rate in bulk PCL with an independent technique were performed by polarized-light optical microscopy (POM). We used a DMRX POM (Leica, Wetzlar, Germany) equipped with a CCD camera (Motic, Hongkong, China) and a THMS600 hot-stage (Linkam, Tadworth, United Kingdom). The sample, which was delivered as a powder, was directly placed between two Linkam circular glass slides and melted by heating to 348 K. At this temperature, a thin film of about 10 µm thickness formed, assisted by slight manual pressing. Further experimental details are provided below.

## 3. Results and Discussion

Figure 2 shows AFM-amplitude images of PCL collected at 330 K after tip-induced crystal nucleation and crystallization at identical temperature. Nucleation at 330 K occurred immediately after having reached the sample surface with the tip. Figure 2A–C were collected after 600, 1800, and 3000 s, respectively, requiring a scanning time of 1200 s for each image. Scanning was performed line-by-line, starting in the top left corner. The line-like indentation pattern of the first contact of the tip with the sample surface for nucleation is visible near the center of the images, indicated with an arrow in Figure 2A. Regarding the effect of crystallization time, the images indicate that primary crystallization (as judged by space-filling) seems finished already before starting scanning Figure 2A. However, the increased amplitude-contrast of Figure 2B,C may suggest further structural changes. For the purpose of clarity, we selected images showing the semicrystalline morphology most clear, being amplitude images in the case of Figure 2; regarding phase images, we observed difficulties due to the non-flat sample surface, complicating the z-axis scaling.

It is important to note that without tip-induced nucleation, the estimated half-time of bulk hot-crystallization of the melt at 330 K is around 10^6^ s [32], with this large value likely related to the absence of active nuclei of any kind. In other words, without a specific nucleation event, crystallization cannot be expected to occur within reasonable time frames. However, the images of Figure 2 reveal that the entire sample crystallized during the collection of the first AFM image, that is, within 1800 s, or even much faster as details of the final morphology are already visible after the first line scans. In other words, the growth rate must be of the order of magnitude of 10^−2^ µm/s, being not in accord with the POM-hotstage-crystallization experiment performed. To measure the bulk crystal-growth rate, the micrometer-sized molten POM sample was in a first step cooled from 348 K at a rate of 5 K/min to 323 K, to permit nucleation and the formation of a small spherulite within 2 min. Subsequently, the sample was heated to 330 K, to monitor the spherulite growth at this temperature. The transfer of the tiny spherulites from 323 to 330 K was performed at a rather low rate of 5 K/min, with an additional isothermal in-between annealing step at 328 K for 7 min, to avoid melting of the small spherulites. Additional growth experiments were performed at 326 and 328 K. Figure 3A shows three POM images of such a PCL spherulite during its growth at 330 K within 20 h. Coloring of the spherulite was obtained by inserting a lambda-retardation plate at a 45°-angle into the light path, providing information about the orientation of the lamellae, being parallel to the spherulite radius, see also [43]. Figure 3B shows spherulite growth rates as a function of temperature, with data obtained on samples of different molar mass taken from the literature (black/gray symbols) [44], or determined in this work (red/gray squares). The data provide information that the growth rate of the specific PCL used here decreases from 4 × 10^−3^ µm/s at 326 K to 2 × 10^−4^ µm/s at 330 K, which, however, is not consistent with the AFM observation. The growth rate at the surface of the sub-micron thin film of the AFM sample is distinctly higher than the bulk growth rate, which may be attributed to reduced constraint of crystal growth at the surface, due to higher mobility of chain segments [43,45,46,47,48,49,50]. In addition, bulk crystallization, as illustrated with the POM images, do not reveal any sign of banding as observed in the AFM image, further highlighted below with Figure 4.

More information regarding the morphology and superstructure can be obtained by inspection of AFM-height images. Figure 4 shows a height image as a pseudo-three-dimensional (3D) representation, collected after crystallization for 3600 s, thus corresponding to Figure 2C. The image suggests that crystallization started at the surface nucleus (see the bright spot/red arrow in the top right part, being ellipsoidal due to the non-perfect line-like tip-contact for nucleation) and led to the formation of a single spherulite within the FSC sample, as concluded from the radial symmetry of the observed morphology. The height profile near the nucleus, along the red line, is not clear, being perhaps caused by the retraction of the tip during the initial nucleation event, pulling out small amount of material. The ring-like structures are interpreted as banding [43,51,52,53,54,55], that is, twisting/rotation of (imaging-wise non-resolved) lamellae when growing parallel to the radius of the spherulite, with the band period seemingly being less than 1 µm [55,56].

As introduced above, thin films of PCL can form flat-on oriented lozenge-shaped lamellae when crystallizing at low supercooling [36]. Recently, Woo’s group [54,55] found that thin films of PCL can form banded spherulites at low supercooling when diluted by poly(phenyl methacrylate) or poly(ethylene oxide). Also, spiral banded spherulites were observed on crystallization between 35 °C and 50 °C, when adding 20% in mass poly(phenyl methacrylate) for dilution [55]. In the present study, the crystallization temperature is 330 K (about 57 °C), causing lower viscosity and molecular interactions compared to lower crystallization temperatures. This might produce a similar effect as diluting the system using a second component.

Finally, using the advantage of the FSC-AFM combination, after annealing/crystallization for 4200 s at 330 K, the sample was heated to 400 K at 10,000 K/s, for proving that crystallization occurred and for comparing the temperature of melting of the formed (surface) crystals with that of crystals grown in bulk. Figure 5 shows with the red curve the corresponding FSC heating scan of the tip-nucleated sample and with the black curve the heating scan of a sample which was crystallized without an initial external nucleation event, presumably nucleated in the bulk. Due to the slow kinetics of bulk nucleation, the crystallization time was increased to 10 days. The melting temperatures of both preparations are equal, which is expected since the crystallization temperatures were identical. As such, it is straightforward to assume a similar thickness of lamellae, controlling the melting temperature according to the Gibbs-Thomson equation [57]. Note, however, that the observed melting temperatures are higher than *T*_m,0_, being not discussed due to their shift to higher temperatures caused by instrumental thermal lag.

The integrals of the curves shown in Figure 5 yield crystallinities of about 20% for both samples. This value is reached after a few thousand seconds after tip-induced surface nucleation but only after 10 days for bulk nucleation. While surface crystallization is complete/area filling after latest 1800 s, bulk crystallization does not result in a volume-filling morphology at the same crystallinity, again highlighting the different growth mechanisms between bulk and surface-nuclei induced crystallization. The annealing time of 10 days roughly corresponds to the half time of crystallization at 330 K [32]. In other words, about half of the sample is filled with spherulitic structures. For the tip-induced surface nucleated sample, the same melting enthalpy/crystallinity implies that half of the sample is filled with crystals, similar to the ones shown in the AFM images of Figure 2 and Figure 4. With the approximate dimensions of 26 × 16 µm^2^ (area) and 0.75 µm (maximum thickness) of the sample given above, the thickness of a crystalline surface layer of the ellipsoid occupying half of its volume can be roughly estimated as 350 nm. In this 350 nm thick surface layer, the crystal growth rate is at least two orders of magnitude faster than in bulk.

## 4. Conclusions

An atomic force microscope (AFM) coupled to a fast scanning chip calorimeter (FSC) has been employed to study AFM-tip induced crystal surface nucleation/crystallization in poly (ε-caprolactone) (PCL) at low melt-supercooling. It was found that the crystal growth rate at the surface is more than two orders of magnitude higher than in the case of bulk crystal growth. While a single spherulite grew with a rate of around 10 µm/600 s (being the distance from the nucleation point to the top left corner in Figure 2A divided by the dwell time before scanning), the bulk crystal growth rate, as determined by classical hot stage microscopy, is around 2 × 10^−4^ µm/s. The higher crystal growth rate of the surface-nucleated sample supports the notion that the mobility of amorphous chain segments at the surface is enhanced compared to the bulk.

## Figures and Tables

**Figure 1 polymers-13-02008-f001:**
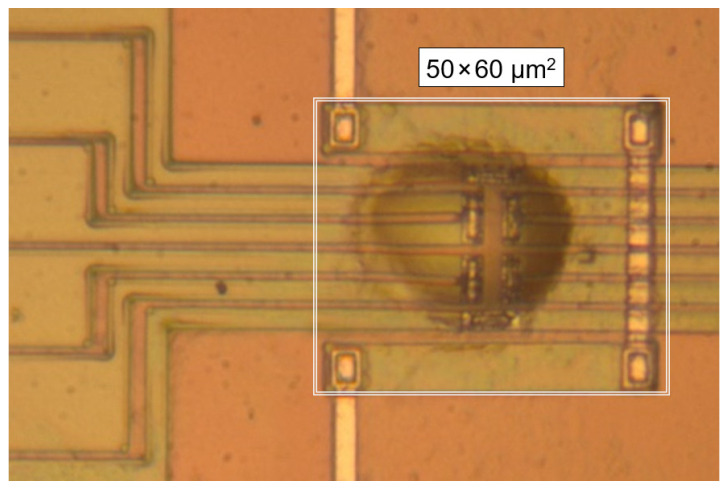
Optical micrograph of the PCL sample on the XI-395 chip-sensor.

**Figure 2 polymers-13-02008-f002:**
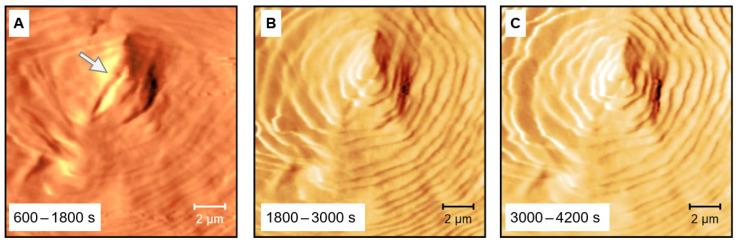
AFM-amplitude images of PCL collected at 330 K after tip-induced crystal nucleation and crystal growth at identical temperature. The arrow in image (**A**) indicates the positions of the first contact of the tip with the sample surface. Figure 2**A**–**C** were collected after 600, 1800, and 3000 s, respectively, requiring a scanning time of 1200 s each.

**Figure 3 polymers-13-02008-f003:**
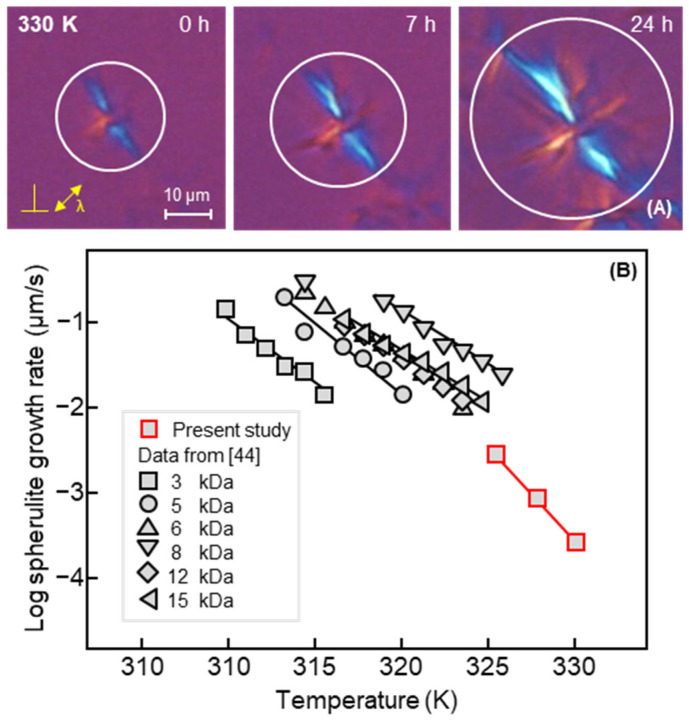
POM images of a PCL spherulite growing at 330 K, collected at the indicated times, serving for estimating the crystal growth rate (**A**). Note that time ‘0 h’ represents the time when the system reached the growth-temperature. The polarizer and analyzer directions as well as the orientation of the long axis of the optic indicatrix of the inserted λ-plate (n_γ_) are indicated in the left image. Spherulite growth rate of PCL as function of temperature (**B**). Data used for comparison are taken from [44], (with permission from John Wiley and Sons, 2007). Note that data of the present work were measured triplicate, with the error bar smaller than the symbol size.

**Figure 4 polymers-13-02008-f004:**
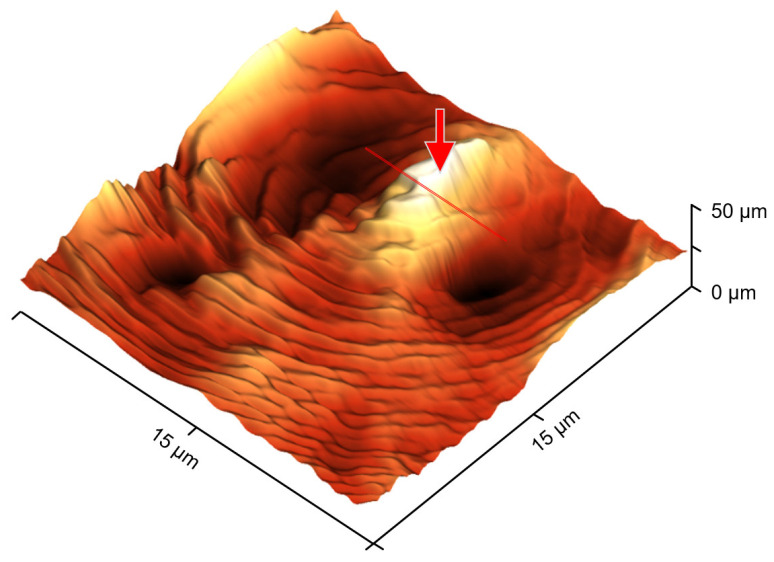
Pseudo 3D-AFM height image of PCL after isothermal crystallization at 330 K for 3600 s (scanning time 3000 to 4200 s).

**Figure 5 polymers-13-02008-f005:**
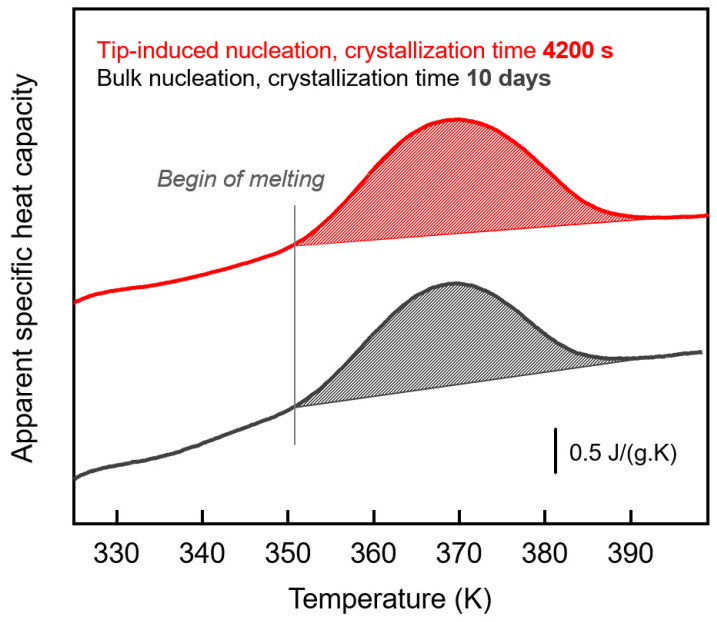
FSC heating scans of PCL, crystallized at 330 K, initiated by AFM-tip-induced surface-nucleation (crystallization time 4200 s) (red, top curve), and by bulk nucleation (crystallization time 10 days) (black, bottom curve). Note that the melting event apparently is observed at a temperature higher than *T*_m,0_, because of instrumental thermal lag at fast heating.

## Data Availability

Data are available on request.

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
