# Peer review of "Surface Crystal Nucleation and Growth in Poly (ε-caprolactone): Atomic Force Microscopy Combined with Fast Scanning Chip Calorimetry"

_polymers, 2021, doi:10.3390/polym13122008_

Round 1

Reviewer 1 Report

The manuscript is very interesting a fit well the scope of the journal. However, it requieres some improvements before publishing.

The state of the art regarding the polymeric matrix (PCL) should be extended in the introducciton section. For instance, PCL is widelly used for the development of materials with shape-memory. Considering that in the present work the crystallization, microstructural and mechancial behaviour are discused, some aspect of shape-memory can be explained.

Minor comments:

- Why Tetrahydrofuran has been selected as solvent? 

Author Response

We want to thank the reviewers for their positive comments, efforts, and detailed suggestions for improving our manuscript. In the following, the comments by the reviewers are marked in normal font, and the responses to the corresponding comments are marked in blue font. Changes in the manuscript are highlighted in yellow.

Reviewer 1

The state of the art regarding the polymeric matrix (PCL) should be extended in the introduction section. For instance, PCL is widely used for the development of materials with shape-memory. Considering that in the present work the crystallization, microstructural and mechanical behavior are discussed, some aspect of shape-memory can be explained.

Though the mechanical behavior of PCL is not in foreground in our study, we agree with the Reviewer that highlighting the importance of PCL as being a superior shape-memory polymer is appropriate. We added this information in the introduction part on line 80, including a reference: Dolynchuk, O., Kolesov, I., Jehnichen, D., Reuter, U., Radusch, H. J., & Sommer, J. U. (2017). Reversible shape-memory effect in cross-linked linear poly (ε-caprolactone) under stress and stress-free conditions. Macromolecules 50(10), 3841-3854.

Why Tetrahydrofuran has been selected as solvent?

Tetrahydrofuran has been proven to be an excellent solvent for PCL as being used for GPC analyses as well as it is not affecting the chemical structure of the polymer.

A corresponding statement was added on lines 107-109.

Reviewer 2 Report

The combination of atomic force microscopy and fast scanning calorimeter (FSC) is interesting and novel and can give interesting information about investigated materials. In the present work the combination of these two techniques is used to study the crystallization of PCL. The obtained by authors results are interesting and should be publish after mayor revisión.

Mayor revisión:

  1. The interesting point is that the AFM image change drastically form Figure 2A to Figure 2B, How you was able to take the AFM image (1200s)? The image was stable during the scanning time?
  2. In the Experimental part you mentioned that you scaned simultaniously height, phase and amplitude AFM images. Why you present only amplitude AFM images? What about height, phase AFM images?

Minor revisión:

  1. Line 105, page 3: please use “tetrahydrofuran” instead of “Tetrahydrofuran”. You do not need a capital letter for solvent name.

Author Response

We want to thank the reviewers for their positive comments, efforts, and detailed suggestions for improving our manuscript. In the following, the comments by the reviewers are marked in normal font, and the responses to the corresponding comments are marked in blue font. Changes in the manuscript are highlighted in yellow.

Reviewer 2

The interesting point is that the AFM image change drastically form Figure 2A to Figure 2B, How you was able to take the AFM image (1200s)? The image was stable during the scanning time?

We do not see that there is a qualitative change of the image/structure when going from Figure 2A to 2B. The contrast is somewhat increased, however, the morphology is settled from the very begin of scanning Image 2A. We even assume that crystallization is finished before the first scan (2A), thus the structure being “AFM-stable”. We do not know the exact reason for the different contrast, when comparing Figures 2A and B/C. It may be related to further minor structural changes.

Wording was changed on line 158.

In the Experimental part you mentioned that you scanned simultaneously height, phase and amplitude AFM images. Why you present only amplitude AFM images? What about height, phase AFM images?

For the purpose of clarity we selected images showing the semicrystalline morphology most clear, being amplitude images in case of Figure 2; regarding phase images, we observed difficulties due to the non-flat sample surface, complicating z-axis scaling. A corresponding statement was added on lines 159-162.

In Figure 4, we used the height contrast, in order to emphasize that the ring-like structures in Figure 2 originate from height oscillations/undulation but not spiral-like growth.

Line 105, page 3: please use “tetrahydrofuran” instead of “Tetrahydrofuran”. You do not need a capital letter for solvent name.

We agree and changed.

Reviewer 3 Report

The presented manuscript describes the synergic combination of AFM with a fast scanning chip calorimeter and follows the surface crystal nucleation/crystallization of poly (ε-rolactone). The manuscript is clearly written and contains new information obtained by meticulous methodology based in the experience of the authors that allowed them to improve the above device.  

I would like to recommend the manuscript for publication in Polymers as it is.

Author Response

We want to thank the reviewers for their positive comments, efforts, and detailed suggestions for improving our manuscript. In the following, the comments by the reviewers are marked in normal font, and the responses to the corresponding comments are marked in blue font. Changes in the manuscript are highlighted in yellow.

Reviewer 3

No changes requested

Round 2

Reviewer 2 Report

publish as is